# What Sustains Mask-Wearing Behavior among Elders in a Rural Community in the Post-COVID-19 Era: An Exploratory Mixed-Methods Study

**DOI:** 10.3390/bs13080678

**Published:** 2023-08-11

**Authors:** Sethapong Lertsakulbunlue, Pinyada Kittisarapong, Sirikorn Pikulkaew, Pree Pusayapaibul, Apisit Tangtongsoonthorn, Chanunpisut Wichaiboon, Fasai Amornchatchawankul, Suranuch Marsook, Supakrit Mahaisawariya, Nattasit Subwongcharoen, Phitchayut Petcharat, Bannawit Luksanasup, Thakornphong Lortharaprasert, Bavorn Tieantanyatip, Anupong Kantiwong, Kanlaya Jongcherdchootrakul

**Affiliations:** 1Department of Pharmacology, Phramongkutklao College of Medicine, Bangkok 10400, Thailand; sethapong.ler@pcm.ac.th (S.L.); anupong.kan@pcm.ac.th (A.K.); 2Medical Cadet, Phramongkutklao College of Medicine, Bangkok 10400, Thailand; pinyadak@pcm.ac.th (P.K.); sirikornp@pcm.ac.th (S.P.); elitenokhuk@gmail.com (P.P.); apisitt@pcm.ac.th (A.T.); chanunpisutw@pcm.ac.th (C.W.); fasaia@pcm.ac.th (F.A.); suranuchm@pcm.ac.th (S.M.); supakritm@pcm.ac.th (S.M.); nattasits@pcm.ac.th (N.S.); phitchayutp@pcm.ac.th (P.P.); bannawitl@pcm.ac.th (B.L.); thakornphongl@pcm.ac.th (T.L.); bavornt@pcm.ac.th (B.T.); 3Department of Military and Community Medicine, Phramongkutklao College of Medicine, Bangkok 10400, Thailand

**Keywords:** COVID-19, face masks, association factors, behavior, elderly, community, rural, Thailand

## Abstract

The current study investigates the factors influencing face-mask-wearing practices among elderly individuals in rural Thailand. A mixed-methods approach was employed, involving qualitative interviews with 15 elderly participants and a subsequent survey of 201 elders. Seven subthemes were identified, including the perceived benefits of mask-wearing, the perceived threat of COVID-19, mask-wearing enhancing attractiveness and self-confidence, social norms, misconceptions about COVID-19 prevention tools, perceived barriers to mask-wearing, and resources to afford face masks. The developed themes, codes, and quotes were utilized for creating a questionnaire. The survey revealed the adherence of 81.1% of the participants to mask-wearing. Structural equation modeling (SEM) analysis demonstrated that motivation, comprising (1) the perceived threat of COVID-19, (2) alternative threats aside from COVID-19, and (3) the perceived benefits of a face mask strongly affected mask-wearing practices (β = 0.68, *p* < 0.001) and the willingness to wear a face mask (β = 0.61, *p* < 0.001). Social norms had a negative direct effect on the perceived barrier (β = −0.48, *p* < 0.001) and a positive direct effect on mask-wearing practices (β = 0.25, *p* = 0.001). This study highlights that motivation and social norms play pivotal roles in sustaining mask-wearing behavior among rural elderly populations. Encouraging local cooperative actions through community rules could initiate behavioral changes within the community. These findings contribute to the understanding of factors influencing mask-wearing and provide insights into designing effective interventions to promote mask-wearing among elderly individuals in rural areas.

## 1. Introduction

Since the COVID-19 pandemic, face masks have played a vital role globally in preventing the spread of the disease [1]. Epidemiologic evidence demonstrates that mask use is 50 to 70% effective in preventing the spread of COVID-19 within a community [2]. Furthermore, surgical masks are known to inhibit PM2.5, which is a major health-related problem in Thailand [3,4,5]. However, the Thai government announced in October 2022 that wearing face masks was no longer considered mandatory, allowing individuals to choose whether to wear masks. Although face masks are no longer required, WHO still recommends that those with a high risk of becoming seriously ill with COVID-19 or dying. For instance, those 60 years of age and older should still wear masks in public [6].

During the pandemic, the Thai population embraced the use of face masks to prevent viral transmission in both symptomatic and asymptomatic individuals [7]. Nevertheless, the mask-wearing behavior of older adults in Thailand is subpar. An online survey conducted in Thailand in 2021 revealed that only 43% of 1230 adults aged at least 60 years from nine provinces across the country’s five regions possessed adequate knowledge of the disease, and only 33% engaged in preventative behaviors [8]. Furthermore, a recent large survey in Thailand demonstrated that those over the age of 60 had a higher proportion of not wearing a face mask outside the home in comparison with those aged 20 to 49 years [9]; thus, more attention should be paid to older adults’ mask-wearing behaviors.

Related research has investigated the factors influencing mask-wearing habits, such as sex, age, level of education, COVID-19 knowledge level, cigarette smoking, affordability, and occupation [8,9,10,11]. Moreover, a recent study regarding the public’s willingness to wear face masks based on the theory of planned behavior revealed that attitudes, social norms, risk perceptions of the pandemic, and perceived benefits of face masks positively influenced the public’s willingness to wear face masks, whereas cost and lack of availability had the opposite effect [12]. Moreover, a study conducted in the US based on the belief–attitude–intention theory demonstrated that physical and communication discomfort had a significant negative effect on the intention to continue wearing a mask [13].

Numerous studies have investigated the various factors influencing mask-wearing, including demographic factors and individual beliefs [2,7,8,9,11,12]. However, it is worth noting that previous research conducted in Thailand has primarily focused on demographic data and has been limited to urban areas. The present study takes a unique approach by employing an exploratory sequential mixed-methods design to examine the influencing factors and beliefs associated with mask-wearing among elderly individuals residing in a rural community in Thailand during the postpandemic era. The findings of this study have the potential to significantly contribute to developing and implementing future behavioral interventions targeting the elderly population in communities across Thailand. By gaining a deeper understanding of the specific factors and beliefs influencing mask-wearing behavior in this context, interventions can be tailored to effectively promote and sustain mask-wearing practices among elderly individuals in rural areas. This research addresses an important gap in the existing literature and offers valuable insights for public health practitioners and policymakers seeking to improve mask-wearing compliance among the elderly in Thailand and similar settings.

## 2. Materials and Methods

### 2.1. Study Design and Subjects

An exploratory sequential mixed-method qualitative–quantitative study was conducted to explore the specific influencing factors for participants residing in rural areas on their mask-wearing behaviors. Our study site was the border area of 4 of 19 villages in Phraphlong Subdistrict, Khao Chakan District, at the border area of Sa Kaeo Province, Thailand, 190 km east of Bangkok [14]. This area has been teaching community medicine for Phramongkutklao College of Medicine since 2002 [14]. The current study was divided into three phases. The initial phase aimed to retrieve possible reasons for mask-wearing in the community using an in-depth interview technique. In the second phase, a questionnaire was designed based on codes and themes from the prior phase. Then, the questionnaire’s reliability and validity were tested. Lastly, the questionnaire was utilized for a cross-sectional study to assess the factors influencing the participants’ mask use.

#### 2.1.1. Qualitative Phase

An in-depth interview was conducted to investigate specific factors influencing mask-wearing behavior in March 2023. The investigator invited elderly participants above 60 years of age using a village health volunteer. Purposive sampling was utilized for selecting 12 participants to conduct in-depth, semi-structured interviews through Zoom, Line video call application, or by phone. Later, face-to-face, in-depth interviews were conducted with three more participants, totaling 15 participants, in order to confirm the thematic saturation (information redundancy occurs, and no new themes or codes ‘emerge’ from the data) [15].

#### 2.1.2. Questionnaire’s Development Phase

In order to develop the questionnaire, a deductive–inductive approach was employed. The items assessing the influencing factors were generated from the extracted codes during the qualitative phase, and the items regarding the willingness and practice of mask-wearing were completed by adapting items from the literature review [12,13,16]. Next, the content validity was assessed. Five professors from the Phramongkutklao College of Medicine (PCM) were asked to comment on the questionnaire’s grammar, word choice, and where phrases should be located. Three items were removed, one for having similar meanings to others and two for being inappropriate for an elderly population. After adjusting the questionnaire, a total of 33 questions were launched on the researcher’s Parents’ Line group platform, and 142 people participated in the online pilot survey. Overall, Cronbach’s alpha scores for the pilot study were at 0.85.

#### 2.1.3. Quantitative Phase

After finalizing the questionnaire, a cross-sectional study was carried out to identify the prevalence and factors associated with mask-wearing among the elderly from 11 March to 15 March 2023. A total of 201 elderly individuals aged 60 or older residing in the study area agreed to participate. The flow of the study is shown in Figure 1.

Exploratory factor analysis (EFA) was later employed to explore the domains of the questionnaire and assess the adequacy of the sample size. The questionnaire was divided into nine domains with an eigenvalue of over 1.0. Four questions with loading factors below 0.4 were removed from the questionnaire, except those within the “perceived threat” domain [17]. Finally, the questionnaire was back-translated and checked by three professors from PCM (Appendix A demonstrates the EFA of the extracted questionnaire).

### 2.2. Data Collection

#### 2.2.1. Qualitative Phase

We first introduced our project to the village health volunteers and have kept in touch since December. After obtaining the ethical approval, data were collected in March 2023 through in-depth, semi-structured interviews. The interviews were carried out with questions and probes to cover the general knowledge of COVID-19, COVID-19 prevention measures, mask-wearing behaviors, and what facilitates and obstructs them. The interviewers were trained at the Phramongkutklao College of Medicine before conducting qualitative research. Preceding the survey, informed consent was obtained. The interviews took approximately 30 to 50 min each and were conducted in Thai; however, a common local dialect included Isan and central dialects. Therefore, the interviews were conducted by researchers fluent in all dialects (PK, CW, TL, NS, PP, FA, SrM, and SpM). The data were collected until the content was saturated, and a voice recorder was utilized for recording and transcribing the conversations. Before conducting the analysis, two researchers (SL and KJ) reviewed the transcription to check for errors.

#### 2.2.2. Quantitative Phase

The investigators informed the village health volunteers before the visiting period (11–15 March 2023) and had face-to-face interviews with the participants. Furthermore, information sheets, objectives, and study methods were provided to the participants. During the study, the questionnaires were self-administered with help from the village health volunteer or the investigators and took approximately 20 to 30 min to complete. In order to de-identify the volunteers, a unique identification number was used in place of their names and identities.

The developed questionnaires consisted of four parts: (1) demographic characteristics (12 items); (2) ten true or false questions about knowledge regarding COVID-19 adapted from the Department of Disease Control, Ministry of Public Health of Thailand, postpandemic knowledge for citizens (10 items) [18]; (3) mask-wearing practice in the past month (4 items); and (4) the questionnaire developed from the qualitative study (29 items). After using EFA to determine the domains of the questionnaire and removing items with loading factors below 0.40, the overall Cronbach’s alpha of the questionnaire was 0.83 and ranged from 0.58 to 0.84 in each domain. The lowest Cronbach’s alpha is 0.58 in the “perceived threat” domain. However, we chose to leave it as-is due to its high corrected item total correlation of above 0.25 [19].

According to the literature review, questions regarding mask-wearing adherence and mask-wearing practice were defined [12,16]. Those who always wore a face mask outside their home or when meeting others were classified as having good adherence, while those who sometimes wore a face mask outside their home or when meeting others and those who did not wear a face mask were classified as having nonadherence. Some of the questions were adjusted to a five-scale score to be included within the structural equation modelling (SEM). The Appendix A contains the full questionnaire used in the present study.

### 2.3. Statistical Analysis

#### 2.3.1. Qualitative Phase

The qualitative investigation employed a team-based, iterative thematic analysis strategy. The interview recordings were transcribed word for word and proofread by the investigators (SL and KJ). All researchers were tasked with familiarizing themselves with the transcripts and then assembled to develop a coding guide based on their initial reading of interview transcripts, which was iteratively revised and refined through team discussion [20]. Inductive and deductive coding were utilized, and the researchers discussed observed patterns and identified key themes. The interpretation of emerging patterns in the data was refined through team discussion. The final findings were then presented in the form of themes and quotations.

#### 2.3.2. Quantitative Phase

Data analyses were performed using StataCorp, 2021, Stata Statistical Software: Release 17. College Station, TX: StataCorp LLC. A frequency distribution of demographic characteristics was performed to describe the study subjects. Categorical data were presented as percentages, and continuous variables were presented as means and standard deviations (SD) or median and interquartile range (IQR) as appropriate. Inferential statistics, including Student’s *t*-test, the Mann–Whitney U test, and Pearson’s chi-squared test, were utilized with a 95% confidence interval (CI). All statistical tests were two-sided, and a *p*-value less than 0.05 was considered statistically significant.

EFA using maximum likelihood extraction and orthogonal (varimax) rotation was performed. Nine domains with eigenvalues greater than one were extracted. The Kaiser–Meyer–Olkin measure for sampling adequacy was applied, yielding an overall index of 0.75, indicating that the data were sufficient for factor analysis. Furthermore, Bartlett’s test for sphericity indicated that the intercorrelation matrix was factorable (χ^2^ = 2633.18, *p* < 0.001). Questions with a loading factor below 0.40 were then removed before being included in the SEM. The variance inflation factor (VIF) was also utilized for assessing multicollinearity between variables. The VIF values ranged from 1.27 to 4.35 and did not exceed 5 [21].

The SEM, using maximum likelihood extraction, was employed to determine how each domain was related and what effect it had on our study population’s willingness to wear a face mask and their mask-wearing practices. The six following indices were utilized for evaluating model fit: (1) the chi-squared test, χ^2^; (2) the chi-squared test over degree of freedom (df), χ^2^/df; (3) the comparative fit index (CFI); (4) the Tucker–Lewis index (TLI), (5) the root-mean square error of approximation (RMSEA); and (6) the standardized root-mean square residual (SRMR). All these indices indicated a proposed fit for SEM data. A χ^2^/df lower than 2, CFI greater than 0.90, TLI greater than 0.90, RMSEA less than 0.08, and SRMR less than 0.08 each indicated an acceptable fit between the data and the hypothesized model [22,23].

## 3. Results

### 3.1. Qualitative Phase

Fifteen elderly participants, consisting of nine males and six females aged between 60 and 72 years, were enrolled to conduct an in-depth interview. Five (33.3%) of these participants exhibited poor mask-wearing adherence (Table 1). Table 2 demonstrates the qualitative findings and compares the extracted questionnaire with the themes and quotes. The themes, subthemes, and codes were also demonstrated in the Appendix A.

Seven subthemes were concluded: four were mask-wearing facilitators, and three were barriers. These seven themes included perceived benefits of mask-wearing, perceived threat of COVID-19, mask-wearing enhancing attractiveness and self-confidence, social norms and new normal, misconceptions about COVID-19 prevention tools, perceived barriers to mask-wearing, and resources to afford face masks (Table 2).

#### 3.1.1. Facilitators/Motivators

Most participants perceived the benefits of mask-wearing for COVID-19 and other respiratory infections. Even though participants believed that the spread of COVID-19 had slowed down in their community, leading them to desire fewer people wearing masks, as they feel their community is safe, they also found masks helpful in protecting against dust and air pollution, especially for agricultural workers exposed to dirt particles. Moreover, participants reported that wearing masks could boost their self-esteem and encourage them to engage in activities. Qualitative participants number 12 (QL12) noted that “When I work in unfamiliar places or go without makeup, wearing a mask gives me more confidence”. Moreover, some villagers used face masks to cover up bad oral hygiene when they had to work early in the morning without brushing their teeth. They believed masks prevented the spread of saliva droplets and food fragments, avoiding potential embarrassment.

Masks are viewed as not always necessary and should primarily be worn in public places and crowded areas, as peer pressure plays a vital role in wearing face masks. Furthermore, elderly villagers have become accustomed to long-term mask use, considering it a habit and no longer bothersome. It is worth noting that role models influence mask-wearing behavior, with some participants mentioning the king’s mask-wearing as a reason to continue wearing masks despite government guidelines (QL15). Others mentioned that they follow their neighbors in mask-wearing decisions.

#### 3.1.2. Barriers/Obstacles

Interestingly, some participants believed that adopting certain measures, such as consuming cooked food, using separate dishware, and maintaining hand hygiene, could reduce the risk of COVID-19 transmission, potentially reducing the need for wearing masks. Additionally, others suggested that using Thai herbal remedies, specifically *Andrographis paniculata* (green chiretta), might suppress COVID-19 and boost immunity, potentially leading to reduced reliance on mask-wearing. One participant shared that using green chiretta had relieved them from the disease, leading others to adopt the remedy as well (QL02).

Participants frequently mentioned using masks as a preventive measure against COVID-19, especially among those working in agriculture. However, some expressed discomfort wearing masks in high-temperature environments, which hinders breathing and working. Moreover, masks were found to obstruct communication and make some elderly individuals feel uneasy.

Most participants believed that the spread of COVID-19 had decreased and suggested reducing mask-wearing in their community, expressing confidence in its safety. Low-income elderly individuals or those without a source of income mentioned the financial burden of purchasing masks during the pandemic. Some participants reused masks for 2–3 days or until they were dirty (QL13), while others chose not to wear masks in places where they perceived COVID-19 as less of a risk. Some opted to buy masks in bulk at a lower price to manage the cost due to limited income (QL14).

### 3.2. Quantitative Phase

The characteristics of participants stratified by mask-wearing adherence are shown in Table 3. A total of 201 elderly individuals living within Phraphlong Subdistrict, Khao Chakan District, Sa Kaeo Province, Thailand, participated in the study. A total of 163 (81.1%) participants always wore a mask outdoors, 28 (13.9%) wore a mask outdoors sometimes, and 10 (5.0%) did not wear a mask. Approximately 60% of the participants were female and aged between 60 and 69 years. Three-quarters of the participants’ highest educational level was primary school. Over 80% of the participants were married and earned an income under THB 5000 monthly. A total of 22.4 and 41.8% smoked and consumed alcohol, respectively. Overall, 69.7% of the participants had used *Andrographis paniculata* to treat or prevent COVID-19. The average score for the ten questions on the COVID-19 quiz was significantly higher among those adhering to wearing a face mask (7.6 ± 1.4) than among those who did not (7.1 ± 1.6) (*p* = 0.027).

Table 4 demonstrates the mean differences in factors influencing mask-wearing among the study population, stratified by domains, after being extracted by the EFA (Appendix A) stratified by mask adherence. Those adhering to mask-wearing had a significantly higher average score of willingness to wear face masks, mask-wearing practice, perceived threat of COVID-19, alternative perceived threat aside from COVID-19, and perceived benefits of face masks in comparison with those not adhering (*p* < 0.001 for all). In addition, the average score of the social norms domain is relatively higher among those adhering to mask-wearing (3.73 ± 1.60) than those not adhering (3.00 ± 1.69) (t = −2.498, *p*-value = 0.007). Surprisingly, those adhering and not adhering to wearing a face mask did not have significantly differing average scores for perceived barriers when wearing a face mask and budget resources for affording face masks (*p* > 0.05). Furthermore, the analysis of each item is demonstrated in the Appendix A.

### 3.3. Structural Equation Modelling

Stata (Version 17.0) was utilized for performing the SEM framework. The goodness of fit was tested, revealing that the normed Chi-square value (χ^2^/df) was 1.52, CFI = 0.91, TLI = 0.90, RMSEA = 0.05, and SRMR = 0.07, indicating a good fit for the data. The SEM is built from 11 latent variables, with mask-wearing as the primary outcome and willingness to wear masks as the secondary outcome (Figure 2). We found that motivation, which is a second-order factor extracted from threat, alternative threat, and perceived benefits, had a strong direct effect on mask-wearing (β = 0.67, *p* < 0.001) and willingness to wear masks (β = 0.47, *p* < 0.001). Social norms had a direct effect (β = 0.25, *p* = 0.001) on mask-wearing practices and a negative direct effect (β = −0.48, *p* < 0.001) on barriers. Surprisingly, barriers, a second-order factor derived from discomfort toward mask-wearing, costs, and misconceptions had a positive direct effect on willingness to wear a mask. Willingness to wear masks also had a direct effect of 0.10 on mask-wearing practices. However, this was without statistical significance. Table 5 demonstrates the SEM result. Finally, the model explained 59% of the variance in mask-wearing practices in the population and 49% of the variance in willingness to wear a mask. The R-squared of the model was higher than the minimum recommended value of 35%, implying a significant interpretation [24].

## 4. Discussion

The present study employed an exploratory sequential mixed-method to identify the factors influencing 201 elders in Phraphlong Subdistrict, Sa Kaeo Province, Thailand. The higher the motivation, the higher the chance of mask-wearing practices and the higher the willingness to wear a face mask. Social norms had a significant effect on mask-wearing practices and a negative effect on mask-wearing barriers, but not on the willingness to wear a face mask.

Other studies conducted during the COVID-19 pandemic have identified perceived barriers, such as cost, mask unavailability, physical discomfort, and communication difficulties, as factors that negatively impact face mask-wearing [12,13]. However, our study revealed that barriers, such as discomfort, costs, and misconceptions, positively affected the willingness to wear a face mask and actual mask-wearing practices. This discrepancy may be attributed to the fact that our study focused exclusively on individuals aged 60 years and above. Older adults tend to perceive initiating behavioral changes as more challenging but find it easier to maintain them once they have been established [25]. Furthermore, although many elderly individuals may find mask-wearing uncomfortable [26], our qualitative study participant (QL07) disclosed that, despite the discomfort, social pressure motivated them to continue wearing a face mask. Another hypothesis suggests that individuals who do not wear face masks may no longer recall the barriers or view them as problematic. However, further investigation is required to understand why motivation positively affects barriers and why barriers positively affect the willingness to wear a face mask.

The prevalence of those who still wear a face mask in this study was 95.0%, and those showing adherence were 81.1%, which is relatively low in comparison with that of a related study because our study was conducted in the post-COVID era. In 2020, a large online survey conducted in Thailand revealed that the prevalence of wearing a face mask outside the home among those aged 60 years and older was 98.9% [9]. A survey based on 708 Malaysian adults also demonstrated that the prevalence of wearing a face mask outside while it was not mandatory was 66.0% [27]. Several studies in Uganda reported that mask-wearing prevalence in rural communities ranged from approximately 33.0 to 70.3% in the later phase of the COVID-19 pandemic [2,28]. Furthermore, the study in Uganda, similar to ours, revealed that knowledge of mask-wearing as a COVID-19 prevention measure was positively associated with mask-wearing adherence [2]. Hence, knowledge of face masks should be promoted, especially among the elderly in rural areas.

Our SEM analysis demonstrated that several factors influenced individuals’ motivation to wear masks. These factors included the perceived threat of COVID-19, the perceived benefits of wearing a face mask, and alternative threats including dust and other respiratory infections. This finding aligns with the health belief model, which posits that behavior is influenced by perceptions of benefits, susceptibility, and severity [29,30]. Previous research also revealed that the use of masks within the community is driven by the fear of contracting COVID-19, particularly among older adults [31,32]. Given that individuals aged 60 years and above are known to be more vulnerable to COVID-19 [6], this fear may be especially pronounced in this age group.

Individuals’ decisions to purchase or use a product are influenced by their optimistic expectations regarding its effects [2]. Thus, the perceived benefits of a face mask contribute to better mask-wearing practices. Face masks were known for their benefits in controlling the COVID-19 spread as well as other respiratory infections [33]. Moreover, face masks are also effective in PM2.5 protection [3]. PM2.5 is a major issue impacting health, both globally and in Thailand [4]. Raising awareness about PM2.5 and the effectiveness of face masks in preventing fine particles might also enhance mask-wearing practices in affected populations.

Similar to our study, previous research also highlighted the significance of social norms in enhancing mask-wearing practices [12]. Within rural communities, individuals are tightly integrated, and, as a result, peers and families strongly influence individuals’ attitudes [29]. Moreover, our findings demonstrate that social norms play a crucial role in reducing barriers to mask-wearing. It is well established that social norms strongly influence individuals’ intentions [34]. Therefore, promoting mask-wearing as a community rule could contribute to adopting better mask-wearing practices, particularly considering that the Thai rural community is primarily governed by community rules established in accordance with the general will and consent of the local population [35].

Based on our qualitative study, the majority of participants no longer perceived COVID-19 to be spreading within their community. However, individuals experiencing upper respiratory symptoms have not yet been tested for COVID-19. Furthermore, the government no longer mandates mask-wearing for all populations, including the elderly, despite the importance of continued mask usage among this group [6]. This lack of enforcement may contribute to reduced disease awareness and the weakening of social measures within the community, which is concerning.

Misconceptions about the use of green chiretta were evident within the community. A significant proportion of participants (over 70%) reported using green chiretta as a treatment or preventive measure for COVID-19, which is relatively high in comparison with the findings from a related study (20%) [36]. While green chiretta has the potential to inhibit viral replication, alleviate symptoms, and reduce the duration and severity of COVID-19 among patients with mild symptoms, its effectiveness in preventing COVID-19 is unsupported [36,37]. Moreover, the inappropriate use of this herb can result in transaminitis and drug-induced liver injury [38,39]. Therefore, it is crucial to provide accurate knowledge to address these misconceptions and educate the community about the potential risks associated with improper use of green chiretta.

The current study found that the perception of the benefits of wearing a face mask and the perceived threat strongly influenced individuals’ willingness and adherence to mask-wearing. In order to promote mask-wearing among local populations, it is vital to cultivate strong local and community leaders who can raise awareness about the threats posed by both COVID-19 and PM2.5 [35]. Furthermore, it is crucial to educate the elderly about the efficacy of face masks in preventing these threats. Additionally, providing knowledge about COVID-19 within the rural community, as well as information on the appropriate use of green chiretta, should be prioritized [2,36,40]. Moreover, considering the findings of this study, the provision of COVID-19 test kits may be essential to facilitate testing for individuals experiencing respiratory symptoms.

The current study encountered several limitations. First, conducting face-to-face interviews results in more socially desirable bias and information bias in comparison with using a fully self-completed questionnaire [41,42]. However, our study population included elderly individuals living in rural areas, and not all the participants knew how to read or write, so assistance was required in answering the questionnaire. In addition, a face-to-face interview would ensure fewer missing data in the study. Second, the study used a cross-sectional survey, which, in turn, makes it difficult to determine the cause-and-effect relationship. However, the study represents the current real-world situation in the postpandemic era of the population. Third, the study only included elders from the central region, so external validation might be necessary. Additionally, the qualitative study was conducted online, potentially missing some details. However, in order to ensure data saturation, three in-depth interviews were conducted face to face. Finally, some of the subjects were elderly and residing in the rural northeast (Isan) areas and were unfamiliar with using the central language. Thus, developing an Isan-dialect questionnaire was preferable. Despite these limitations, our research was the first to study factors influencing mask-wearing among the elderly in a rural population in Thailand. Furthermore, a mixed-methods approach was utilized to enrich our results.

## 5. Conclusions

In conclusion, the current study sheds light on the current state of mask-wearing among elderly individuals residing in remote rural communities in Thailand. Our findings suggest that mask-wearing practices in this population may be inadequate. The key determinants for sustaining mask-wearing behavior among rural elderly individuals were identified as motivation and social norms. Encouraging behavioral changes within the community can be initiated through the implementation of community rules promoting local cooperation. In addition, it is crucial to enhance knowledge regarding COVID-19 prevention measures and the proper utilization of resources, such as *Andrographis paniculata*.

Regulations enforced by law, especially in conditions of health and life-threatening situations, have become ingrained in individuals’ consciousness as a motivational factor and have also been integrated into social norms. By addressing these factors, interventions can effectively promote and maintain mask-wearing behavior among the elderly in rural areas. The implications of the present study extend to public health practitioners and policymakers who aim to develop targeted interventions to improve compliance and promote behavioral change among the elderly in similar remote rural settings.

## Figures and Tables

**Figure 1 behavsci-13-00678-f001:**
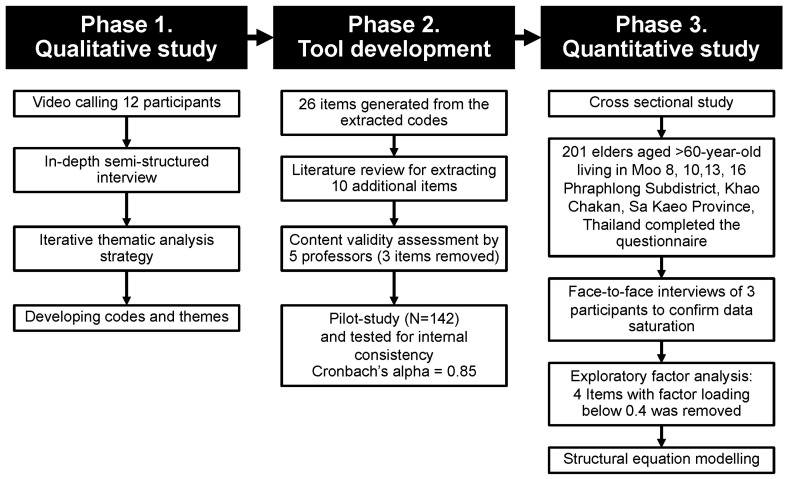
Flow of the study process.

**Figure 2 behavsci-13-00678-f002:**
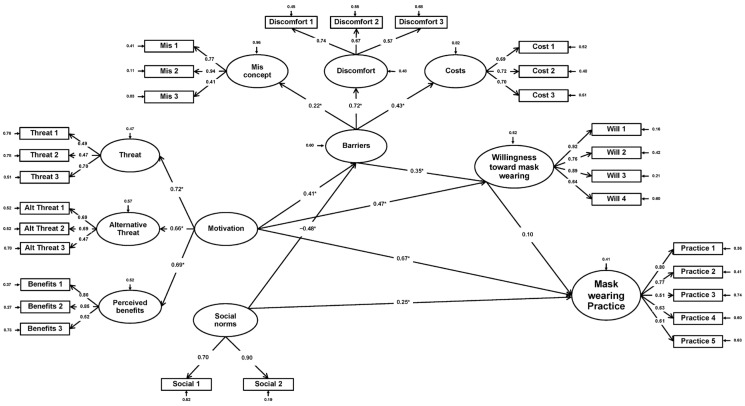
Structural equation model of factors influencing mask-wearing. * *p* < 0.05.

**Table 1 behavsci-13-00678-t001:** Characteristics of qualitative interview participants (N = 15).

Participants Code	Age	Gender	Mask Adherence
QL01	64	Male	Not adhering
QL02	61	Female	Adhering
QL03	65	Female	Adhering
QL04	60	Female	Adhering
QL05	66	Female	Adhering
QL06	64	Male	Adhering
QL07	63	Male	Adhering
QL08	63	Male	Not adhering
QL09	72	Female	Not adhering
QL10	61	Male	Adhering
QL11	64	Male	Adhering
QL12	72	Male	Adhering
QL13	67	Male	Adhering
QL14	65	Female	Not adhering
QL15	63	Male	Not adhering

QL: qualitative participants code.

**Table 2 behavsci-13-00678-t002:** Qualitative findings of factors influencing mask-wearing and questionnaire extraction.

Subthemes	Quotes	Questionnaire Extraction
Facilitators/motivators subthemes		
Perceived benefits of mask-wearing	“It provides good protection, and wearing it feels comfortable. Even though the COVID infection rate has decreased, a mask still helps protect our airways.” QL05	1. I believe in the effectiveness of face masks for COVID-19 protection.
	“I wear a mask for dust protection. Dust is the main reason, as there is road construction and non-asphalt roads. When trucks drive by, dirt particles fly up, so I continue wearing a mask.” QL02	2. I wear a mask to protect myself from air pollution.
	“I am afraid that diseases will come again, not only COVID-19, but also other diseases.” QL08	3. I wear a mask to protect myself against other respiratory infections apart from COVID-19.
	“I believe that mask-wearing will reduce the spreading of COVID-19. It can protect us when we are close; when we talk, our saliva cannot spread out. Masks cover us individually.” QL12	4. I believe that mask-wearing helps reduce the spread of COVID-19.
Perceived threat of COVID-19	“I rate my anxiety related to COVID-19 as being at its lowest. My anxiety has decreased since there have been no new cases in our community.” QL05	5. I believe that COVID-19 is still spreading throughout my community.
	“I am still worried though. The disease may come back again.” QL08	6. I believe that COVID-19 can reemerge in the future.
	Why do you choose to wear it even though it can make breathing difficult? “It’s because I am afraid. Afraid of the disease.” QL07	7. I still feel anxious about being infected with COVID-19.
Mask-wearing enhancing attractiveness and self-confidence	“I’m confident. When I look in the mirror, I think I look beautiful.” QL09	8. I believe that using a face mask makes me look better.
	“Wearing a mask makes me feel more confident.” QL02	9. Wearing a face mask improves my self-esteem.
	“… After eating, conversations often occur. Because we may not have brushed our teeth, we worry that there may be unclean substances in our mouths … Wearing a mask can make us smile brightly.” QL02	10. I wear a face mask to conceal my poor oral hygiene.
Social norms and new-normal	“When I go out, I always wear a mask because if I don’t and I see other people, they might think that I’m weird or that I don’t care about my own health. Therefore, I make sure to bring a mask with me at all times so that I won’t be criticized by anyone.” QL14	11. I am worried about other people’s perspectives when I remove my face mask in public.
	“I am familiar with masks and wearing them has become a habit. It no longer bothers me.” QL09	12. I am familiar with mask-wearing.
	“Not everyone wears masks to the event, but the host provides them for those who forget.” QL03	13. I wear a face mask because it is a social measure.
	“These days, I still see people wearing masks. However, if I am the only one wearing a mask, I will not want to wear it either. But since there are still people wearing masks, being in a group that wears masks is better.” QL07	14. I wear a face mask because others are also wearing them.
	“I don’t want to talk to or warn him (Mask-wearing) because he will accuse me of being disgusted with him. It’s better to keep my distance and let him figure it out on his own. Keeping my distance is the best option. If I were to warn him, he might accuse me of snubbing him.” QL02	15. I feel anxious or disgusted when someone not wearing a mask gets close to me.
Barriers/obstacles subthemes		
Misconceptions about COVID-19 prevention tools	“Don’t be afraid. I have taken green chiretta and been relieved of the disease. Because others believe it, they have also started planting green chiretta.” QL02	16. I trusted that *Andrographis paniculata* could effectively prevent COVID-19, so I reduced my mask-wearing.
	“The COVID-19 virus can be found in uncooked food because animals that we eat may sometimes be infected with the virus. To prevent infection, it’s important to eat cooked food, use serving spoons, and wash your hands regularly.” QL02	17. I trusted that practices such as eating cooked food, using serving spoons, and washing hands could effectively prevent COVID-19, so I reduced my mask-wearing.
	“Wearing a cloth mask can protect me against various things, including dust and germs. It can also be washed and reused multiple times. Both cloth masks and surgical masks are similarly effective in preventing illness caused by germs.” QL11	18. I trusted that cloth face masks could effectively prevent COVID-19, so I reduced my mask-wearing.
Perceived barriers to mask-wearing	“I cannot wear a mask all the time. It feels tight and makes it hard to breathe.” QL07	19. I feel uncomfortable when wearing a mask.
	“It’s hot. I work in the sun, so it’s hot. I can’t wear it.” QL04	20. Wearing a mask impairs my work productivity.
	“It is uncomfortable and makes me feel uneasy. Going out is a challenge because it restricts my ability to speak loudly and clearly, which can make it hard for me to communicate with others.” QL07	21. Wearing a mask can be an obstacle to communication.
	“After wearing it for a while, the material can become irritating, and some children who wear masks develop rashes on their faces.” QL02	22. I think that I am allergic to face masks, e.g., itchy, acne, or rash.
Resources to afford face masks	“The fact that they don’t want to wear it is that the mask they have doesn’t meet the standard.” QL02	23. I do not have enough high-quality face masks to meet the standard.
	“I bought them only for my grandchildren who go to school because masks are expensive, whether they come in a pack or a box.” QL10	24. The high price is affecting my decision to purchase a face mask.
	“If there’s no control over the spread of the disease and contribution of masks, they may not have enough money to purchase them.” QL10	25. I do not have enough money to afford using a face mask daily.
	“Yes, it definitely increases expenses because people have to keep buying more masks when they run out. It’s definitely an added expense.” QL01	26. I think that buying face masks increases my financial burden.

QL: Qualitative participants code.

**Table 3 behavsci-13-00678-t003:** Characteristics of participants by mask-wearing adherence (N = 201).

Characteristics	Adhering	Not Adhering	*p*-Value
	*n* (%)	*n* (%)	
Sex			0.446 ^a^
Male	62 (78.5)	17 (21.5)	
Female	101 (82.8)	21 (17.2)	
Age			
Mean ± SD	68.5 ± 7.8	70.6 ± 8.1	0.113 ^b^
Median (IQR)	69.5 (64.0–77.0)	67 (62.0–73.0)	0.582 ^c^
Age groups			0.479 ^a^
60–69	97 (83.6)	19 (16.4)	
70–79	46 (79.3)	12 (20.7)	
≥80	20 (74.1)	7 (25.9)	
Status			0.833 ^a^
Single	12 (85.7)	2 (14.3)	
Married	126 (80.3)	31 (19.8)	
Divorced/widowed	25 (83.3)	5 (16.7)	
Ethnicity			0.469 ^a^
Thai	161 (81.3)	37 (18.7)	
Laos	2 (66.7)	1 (33.3)	
Educational level			0.679 ^a^
Below primary school	38 (79.2)	10 (20.8)	
Primary school	117 (82.4)	25 (17.6)	
Above primary school	8 (72.7)	3 (27.3)	
Occupation			0.939 ^a^
Unemployed	43 (82.7)	9 (17.3)	
Agriculture	90 (80.4)	22 (19.6)	
Others	30 (81.1)	7 (18.9)	
Scheme			0.439 ^a^
Universal coverage	155 (81.6)	35 (18.4)	
Others	8 (72.7)	3 (27.3)	
Monthly income (THB/month)			0.555 ^a^
<1000	70 (86.4)	11 (13.6)	
1000–4999	64 (77.1)	19 (22.9)	
5000–9999	11 (78.6)	3 (21.4)	
10,000–14,999	7 (77.8)	2 (22.2)	
≥15,000	11 (78.6)	3 (21.4)	
Alcohol drinking			0.697 ^a^
Never	57 (78.1)	16 (21.9)	
Previous drinker	36 (81.8)	8 (18.2)	
Current drinker	70 (83.3)	14 (16.7)	
Smoking status			0.831 ^a^
Not-current smoker	127 (81.4)	29 (18.6)	
Current smoker	36 (80.0)	9 (20.0)	
*Andrographis paniculate* usage			0.565 ^a^
No	48 (78.7)	13 (21.3)	
Yes	115 (82.1)	25 (17.9)	
Score of COVID-19 quiz			0.077 ^a^
Less than 8	77 (76.2)	24 (23.8)	
8 or more	86 (86.0)	14 (14.0)	
Mean ± SD	7.6 ± 1.4	7.1 ± 1.6	0.027 ^b^

^a^ Chi-square test, ^b^ independent Student’s *t*-test, and ^c^ Mann–Whitney U test. SD: standard deviation IQR: interquartile range and THB: Thai Baht.

**Table 4 behavsci-13-00678-t004:** Overall mean differences in factors influencing mask-wearing among elders in a rural community in central Thailand stratified by mask adherence.

Question Domain	Mask Adherence	Mean ± SD	*t*	*p*-Value
Overall willingness to wear face masks—Likert scale	Not adhering	3.07 ± 1.50	−3.610	<0.001
	Adhering	3.97 ± 1.36		
Overall mask-wearing practice—Likert scale	Not adhering	2.93 ± 1.19	−12.331	<0.001
	Adhering	4.41 ± 0.47		
Overall perceived threat—Likert scale	Not adhering	2.25 ± 1.12	−3.106	<0.001
	Adhering	2.97 ± 1.31		
Overall alternative perceived threat aside from COVID-19—Likert scale	Not adhering	2.73 ± 1.57	−3.551	<0.001
	Adhering	3.56 ± 1.22		
Overall misconception in COVID-19 prevention tools—Likert scale	Not adhering	3.76 ± 1.35	0.727	0.234
	Adhering	3.58 ± 1.38		
Overall perceived benefits of face mask—Likert scale	Not adhering	3.21 ± 1.47	−4.193	<0.001
	Adhering	4.08 ± 1.07		
Overall discomfort when wearing a face mask—Likert scale	Not adhering	2.98 ± 1.44	−1.132	0.130
	Adhering	3.27 ± 1.38		
Overall social norms—Likert scale	Not adhering	3.00 ± 1.69	−2.498	0.007
	Adhering	3.73 ± 1.60		
Overall budget resources for affording face masks—Likert scale	Not adhering	3.20 ± 1.44	−0.528	0.299
	Adhering	3.34 ± 1.42		

SD: standard deviation.

**Table 5 behavsci-13-00678-t005:** Structural equation model results.

Independent Variable	Barriers			Willingness			Practice		
Dependent Variable	TE	DE	IE	TE	DE	IE	TE	DE	IE
Motivation	0.41	0.41 **	-	0.61	0.47 ***	0.14 *	0.68	0.67 ***	0.01
Social norms	−0.48	−0.48 ***	-	-	-	-	0.23	0.25 **	−0.02
Barriers	-	-	-	0.35	0.35 **	-	0.04	-	0.04
Willingness	-	-	-	-	-	-	0.10	0.10	-
R-squared	0.40			0.49			0.59		
χ^2^/df = 1.52, CFI = 0.91, TLI = 0.90, RMSEA = 0.05, and SRMR = 0.07.									

TE: total effect, DE: direct effect, IE: indirect effect, CFI: Comparative fit index, TLI: Tucker–Lewis index, RMSEA: Root Mean Square Error of Approximation, and SRMR: Standardized Root Mean Squared Residual. * *p* < 0.05; ** *p* < 0.01; *** *p* < 0.001.

## Data Availability

The datasets generated and/or analyzed during the current study are not publicly available because the dataset contains sensitive identifying information. Due to the in-place ethical restrictions, the datasets are available from the author upon reasonable request (contact Sethapong Lertsakulbunlue via sethapong.ler@pcm.ac.th).

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
