# Peer review of "What Sustains Mask-Wearing Behavior among Elders in a Rural Community in the Post-COVID-19 Era: An Exploratory Mixed-Methods Study"

_behavsci, 2023, doi:10.3390/bs13080678_

Round 1
Reviewer 1 Report
the article presents compelling data and draws valuable conclusions, I believe that incorporating the powerful data visualization capabilities offered by RStudio could further enhance the presentation of results. RStudio, as a widely used integrated development environment for statistical computing and graphics, could assist in improving the clarity and impact of the visual representation of the data.
By utilizing RStudio, the authors could explore various graphing techniques, such as the creation of visually appealing and informative plots, charts, and tables. RStudio's extensive library of packages, such as ggplot2 and shiny, could facilitate the production of dynamic and interactive visualizations. These enhancements would aid readers in better understanding the patterns and trends in mask usage adherence, thus reinforcing the significance of the findings.
Furthermore, RStudio provides an array of statistical tools that could be applied to further analyze the data presented in the article. Techniques such as regression analysis or time series modeling could provide additional insights and support the authors' conclusions. Additionally, the inclusion of statistical measures and confidence intervals in the tables would help quantify the significance of the observed effects.
In conclusion, I encourage the authors to consider leveraging RStudio's capabilities to improve the visualization and statistical analysis of their data. This would undoubtedly strengthen the impact of their research and facilitate a more comprehensive understanding of the factors influencing mask usage adherence. I commend the authors for their valuable contribution to the field and eagerly await further advancements in this critical area of study.
need a native english speaker to read and correct the manuscript
Author Response
Thank you for your kind suggestions. However, the study focuses on the exploratory mixed-method and the structural equation model (SEM) developed using the STATA software (StataCorp, 2021, Stata Statistical Software: Release 17. College Station, TX: StataCorp LLC.), which may limit further visualization of the SEM using RStudio. We believe that the SEM provided a comprehensive analysis of the available data. Additionally, we have improved the presentation of the data based on various suggestions from other reviewers. I truly appreciate your kind words and will consider using RStudio to enhance my future studies.

Reviewer 2 Report
This study aimed at understanding the factors influencing mask-wearing practices among elderly individuals in rural Thailand. The focus on elderly individuals in rural areas is an important contribution, as rural populations may face unique challenges and barriers in adopting preventive behaviors such as mask-wearing. The study's findings have direct implications for designing interventions tailored to rural communities. The combination of qualitative interviews and a subsequent survey allows for a comprehensive exploration of the topic, capturing both in-depth insights from qualitative data and broader patterns from quantitative data. The used thematic analysis added depth and richness to the findings, allowing for a nuanced understanding of the participants' perspectives. The use of SEM gave a better view on the complex relationships among variables and provided statistical evidence for the identified factors influencing mask-wearing practices. Overall, these strengths enhance the study's credibility and contribute to the understanding of factors influencing mask-wearing practices among elderly individuals in rural Thailand.
Some minor issues: 1) how was the sample size for both phases determined? 2) it would be very helpful to all a table describing the characteristics of each of 15 participants. For each participant, include a description for the gender, age, marital status, education, occupation, etc…. Each row can be labeled as Participant 1, participant 2, etc… The results then can refer to those numbers so readers can link quotes with the participant. 3)QL stands for?
Thank you
na
Author Response
Thank you for your kind suggestions, we've amended as commented.

Reviewer 3 Report
The authors focused on the important problem of influencing communities in difficult situations in such a way that the behavior imposed by law was adopted by individuals as motivation and social norms. The analysis was based on behaviors that emerged after the covid-19 pandemic.
The study was carefully planned and conducted. However, there are some doubts about it:
1/ in line 15 it was stated that 15 people took part in the qualitative study, and in line 111 that 12 people took part,
2/ the question arises whether it is not enough to construct a questionnaire based on their answers,
3/ moreover, the interviews were conducted on-line - is this the right method to interview elderly people living in rural areas? Didn't this method lead to the selection of participants of a special nature? Are the answers to the survey representative of people over 60 from agricultural areas?
4/ summarizing the doubts - are the obtained results fully reliable?
However, the phenomenon that has become a research problem is very important. It turns out that regulation enforced by law - the fact is that in conditions of health and life threat - has been built into the consciousness of individuals as a motivational factor, and it has also been included in social norms. This is the most important conclusion of this article. It forms the basis for public health research as well as the basis for health policy. I presume that not only in the field of health care.
Author Response
Thank you for your kind suggestions, we've amended as commented.
We've responded to the questions on the reliability of the study in the attached file.

Reviewer 4 Report
Introduction:
From a public health perspective, mask-wearing is a crucial issue. Particularly, the study on mask-wearing among elderly individuals in rural areas, rather than urban regions, is of great interest.
Research Methodology:
You followed reasonable procedures to construct the questionnaire.
Statistical Analysis:
Reasonable procedures were followed to perform statistical analysis.
Results:
The results are described in a specific manner, but they should be written in a way that enhances reader comprehension.
Sections 3.1.1, 3.1.2, 3.1.3, 3.1.4, 3.1.5, 3.1.6, 3.1.7:
This section appears to be represented in Table 1. Rather than repeating Table 1, a comprehensive description of the overall content is warranted.
Table 3:
Table 3 provides a highly detailed representation of specific sub-items. However, due to the excessive number of items, it might be better to present the information through scoring based on sub-themes rather than individually.
This study seems to have conducted a highly detailed investigation into mask-wearing among elderly individuals in rural areas. The findings appear to be of significant value to public health. Nonetheless, there are certain aspects of the results that require revision.
Author Response

(The authors gave the same response as above.)

Round 2
Reviewer 1 Report
accepted.